# Careful at Estimation and Bold at Exploration for Deterministic Policy Gradient Algorithm

## Abstract

Exploration strategies within continuous action spaces often adopt heuristic approaches due to the challenge of dealing with an infinite array of possible actions. Previous research has established the advantages of policy-based exploration in the context of deterministic policy reinforcement learning (DPRL) for continuous action spaces. However, policy-based exploration in DPRL presents two notable issues: unguided exploration and exclusive policy, both stemming from the soft policy learning schema, which is famous for DPRL policy learning. In response to these challenges, we introduce a novel approach called Bold Actor Conservative Critic (BACC), which leverages Q-value to guide out-of-distribution exploration. We extend the dynamic Boltzmann softmax update theorem to the double Q function framework, incorporating modified weights and Q values. This extension enables us to derive an exploration policy directly for policy exploration, which is constructed with the modified weights. Furthermore, we explicitly utilize the minimum Q value as an intermediate step in policy gradient computation, which derives from a conservative policy. In practice, we construct such an exploration policy with a limited set of actions and train a parameterized policy by minimizing the expected KL-divergence between the target policy and a policy constructed based on the minimum Q value. To evaluate the effectiveness of our approach, we conduct experiments on the Mujoco and Roboschool benchmarks. Notably, our method excels in the highly complex Humanoid environment, demonstrating its efficacy in tackling challenging continuous action space exploration problems.

## 1 Introduction

Deep reinforcement learning(RL) has attracted much attention in recent years. It has achieved massive success in many fields, such as DQN (Mnih et al., 2015) in simple RGB games, AlphaStar (Vinyals et al., 2019), and OpenaiFive (OpenAI et al., 2019) in multi-player combat games, chatGPT (OpenAI, 2022) in natural language processing. When applying deep RL in continuous action control, such as robotic control, higher demands exist on the robustness of reinforcement learning policy (Haarnoja et al., 2017). Algorithms based on the maximum entropy framework (Ziebart, 2010) are more robust due to the diverse action selection, which augments the standard reward with the policy entropy, to some extent, encourages exploration in training and finally derives a robust policy. The intuitive reason for taking exploratory actions is that other actions with lower predicted rewards may be better. Moreover, the method used to select actions directly affects the rate at which the RL algorithm will converge to an optimal policy. Ideally, the algorithm should perform a non-greedy action if it lacks confidence in the current prediction and perform a bold exploration once we gather more information about the prediction result.

Although various exploration methods, such as $\epsilon$-greedy, Softmax, UCB-1 (Auer et al., 2002), have been suggested for use in discrete action space, these kinds of explorations are not the same thing as the exploration in the continuous action space, due to the infinite actions. Since the actions in continuous space are uncountable, the exploration strategy is heuristic, such as adding the Gaussian perturbation (Silver et al., 2014; Van Hasselt et al., 2016; Fujimoto et al., 2018; Haarnoja et al., 2018). Intuitively, this kind of **unguided exploration** should not be an efficient exploration strategy. It will slow down learning the optimal policy due to its randomness. However, such approaches still

yield significant improvements compared to methods not incorporating exploration. For instance, in the TD3 (Fujimoto et al., 2018) algorithm, random noise is added to actions for exploration purposes. Actions that exceed bounds are clipped to ensure their validity. Then, policy-based noise is considered in the SAC (Haarnoja et al., 2018) algorithm, which also achieved good results compared to the previous method. SAC considers a stochastic actor to explore and minimize the reverse KL-divergence to learn a policy for computation consideration. However, this reverse KL-divergence leads to another issue: **exclusive policy**, which hinders exploring the optimal policy. This issue lies in the Q value-based policy gradient method since the DPG (Silver et al., 2014) algorithm is proposed. That is, policy learning and Q-function learning are separate processes. Policy learning lags behind Q-function learning, meaning that actions collected from the policy have relatively lower Q-values. As we assume the Q function is multimodal, the policy may be sub-optimal due to the exclusive reverse KL-divergence, preferring an unimodal approximation.

To overcome the unguided exploration issue, the OAC (Ciosek et al., 2019) algorithm uses the Q-value-based policy gradient to predict Gaussian mean offset, then uses offset compensation Gaussian policy to explore. However, in practice, most of these predictions are inaccurate regarding high-dimensional action space, which leads to efficient exploration. A natural and straightforward idea is using Q-value to guide exploration. Suppose the high Q-value actions far from the policy can also be sampled. In this case, the exclusive policy issue can also be avoided. In this paper, we propose **B**old **A**ctor **C**onservative **C**ritic (BACC) algorithm to achieve such Q-value-guided out-of-distribution (OOD) exploration.

Specifically, we initially introduce the DDQS operator based on the Double Q-function framework. This operator is an extension derived from the DBS operator. Within the Double Q-function framework, our primary modifications involve the weights and values associated with computing expected state values of the DBS operator. For given state-action pairs, we softmax the maximum value (greedy Q value) of two Q functions as weights and take the minimum value (conservative Q value) of the two Q functions as values. Subsequently, we provide proof of the convergence of the DDQS operator (Theorem 1), which serves as an assurance of the feasibility of the exploration method we propose. Following this, we extract weights to construct an exploration policy to guide action exploration, which is similar in approach to the SARSA (Rummery & Niranjan, 1994) algorithm. However, considering sample efficiency, we aim to separate action exploration from policy learning. Therefore, we superficially softmax the conservative Q values to construct an optimization policy. We then iteratively minimize the KL-divergence between the target policy and the conservative policy to guide target policy learning. According to Theorem 1, both two Q functions will eventually converge to the optimal Q-function. Therefore, the exploration policy and the optimization policy will also ultimately be the same, which is the optimal policy. Consequently, minimizing the KL-divergence can lead to obtaining the actual optimal policy. The use of conservative Q-values in the optimization policy wasn't our initial contribution, but in this paper, it is the first time it is explicitly related to the conservative policy, to achieve stable policy learning.

We evaluate our proposed method on Mujoco (Todorov et al., 2012) benchmarks and verify that the proposed method outperforms the previous state-of-the-art in various environments, particularly the most complex Humanoid environment. We achieved about 8k scores in 3 million steps, a massive improvement over previous methods. We also tested our method in the Roboschool (OpenAI, 2017) environments HumanoidFlagrun and HumanoidFlagrunHarder. The results indicate that our exploration method is more robust than the OAC algorithm in complex environments.

## 2 PRELIMINARY

We first introduce notation and the maximum entropy objective, then summarize the Soft Policy Learning method.

**Notation.** In this paper, we consider deterministic policy reinforcement learning method for continuous action space. Consider a discounted infinite-horizon Markov decision process (MDP), defined by the tuple $(\mathcal{S}, \mathcal{A}, p, r, \gamma)$, where the state space $\mathcal{S}$ and the action space $\mathcal{A}$ are continuous, and the state transition probability $p : \mathcal{S} \times \mathcal{A} \times \mathcal{S} \to [0, \infty)$ represents the probability density of the next state. Given the state $\mathbf{s}_t \in \mathcal{S}$ and action $\mathbf{a}_t \in \mathcal{A}$ at time-step $t$, we can get the probability density of $\mathbf{s}_{t+1} \in \mathcal{S}$. The environment emits a bounded reward $r : \mathcal{S} \times \mathcal{A} \to [r_{\min}, r_{\max}]$ on for specific state

and action pair. $\gamma$ is the discount factor, and its value is in the range $[0, 1)$, which makes the infinite accumulated reward finite in mathematics.

**Maximum entropy objective.** Standard RL algorithm maximizes the expected sum of rewards $\sum_t \mathbb{E}_{(\mathbf{s}_t, \mathbf{a}_t) \sim \rho_\pi} [r(\mathbf{s}_t, \mathbf{a}_t)]$. $\rho_\pi(\mathbf{s}_t, \mathbf{a}_t)$ denotes state-action marginals of the trajectory distribution induced by a policy $\pi(\mathbf{a}_t | \mathbf{s}_t)$. Maximum entropy objective augment the expectation with the expected entropy of the policy over $\rho_\pi(\mathbf{s}_t)$:

$$J(\pi) = \mathbb{E}_\pi \left[ \sum_{t=0}^{\infty} r(\mathbf{s}_t, \mathbf{a}_t) + \alpha \mathcal{H}(\pi(\cdot | \mathbf{s}_t)) \right].$$

The temperature parameter $\alpha$ balance the relative importance of the entropy term and the reward, and this entropy term influence the exploration of the policy, which in result to a more stochastic optimal policy ideally.

**Soft policy learning.** Soft policy maximizes the maximize entropy objective and modifies the Q value function using the standard Q value function minus the current action's log probability, this Q value is called Soft Q value. Considering the discount factor in practice algorithm, the standard Q value function is $\mathbb{E}_{(\mathbf{s}_t, \mathbf{a}_t) \sim \rho_\pi} [\sum_{t=0}^{\infty} \gamma^t r(\mathbf{s}_t, \mathbf{a}_t)]$. The soft Q value is $\sum_{t=0}^{\infty} \mathbb{E}_{(\mathbf{s}_t, \mathbf{a}_t) \sim \rho_\pi} [\gamma^t r(\mathbf{s}_t, \mathbf{a}_t) + \alpha \gamma^{t+1} \mathcal{H}(\pi(\cdot | \mathbf{s}_{t+1}))]$. For a fixed policy, the soft Q value can be computed iteratively, starting from any function $Q : \mathcal{S} \times \mathcal{A} \to \mathbb{R}$ and repeatedly applying the modified Bellman backup operator $\mathcal{T}^\pi$ given by

$$\mathcal{T}^\pi Q(\mathbf{s}_t, \mathbf{a}_t) \triangleq r(\mathbf{s}_t, \mathbf{a}_t) + \gamma \mathbb{E}_{(\mathbf{s}_{t+1}, \mathbf{a}_{t+1}) \sim \rho_\pi} [Q(\mathbf{s}_{t+1}, \mathbf{a}_{t+1}) - \alpha \log \pi(\mathbf{a}_{t+1} | \mathbf{s}_{t+1})],$$

then improve the policy by minimizing following formula

$$\pi' = \arg \min_{\pi \in \Pi} D_{KL} \left( \pi(\cdot | \mathbf{s}_t) \,\Big\|\, \frac{\exp(Q(\mathbf{s}_t, \cdot))}{Z(\mathbf{s}_t)} \right),$$

where $Z(\mathbf{s}_t) = \sum_{\mathbf{a}_t} Q(\mathbf{s}_t, \mathbf{a}_t)$ normalizes the distribution.

## 3 PROBLEMS AND OUR SOLUTION TO PREVIOUS WORK

In this section, we provide illustrations to explain inefficient exploration. Then, we demonstrate why the method proposed in this paper is effective. For a fixed state, the Q-values and policy regarding one-dimensional actions are approximately as shown in Fig. 1. It is typically assumed that the Q-function is multimodal, and the policy is modeled as a Gaussian distribution.

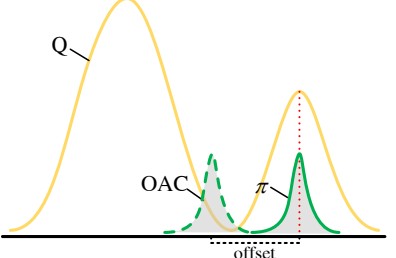

(a) Unguided exploration and exclusive policy

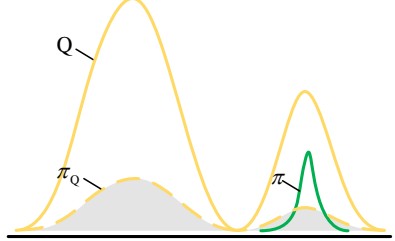

(b) Q-value-guided OOD exploration

Figure 1: Left: Exploration typically occurs around the policy $\pi$. However, due to the exclusiveness of KL-divergence and the delay in policy learning, there is a high likelihood that the policy could become stuck at a suboptimal state. The OAC algorithm predicts an offset to allow better exploration, however, this offset cannot be accurately predicted in high-dimensional action spaces. Right: Exploring with $\pi_Q$, constructed by softmaxing the Q-value, can help avoid sub-optimality. In contrast to the policy $\pi$, it represents a form of out-of-distribution exploration.

**Unguided exploration.** refers to a form of exploration in which an agent takes random actions without a clear goal. This type of exploration can be inefficient and time-consuming, as the agent may spend significant amounts of time exploring unimportant or irrelevant areas of the environment. Exploration that relies solely on the current policy is limited by the quality of the policy initialization and the difficulty of improving the policy.

**Exclusive policy.** occurs in soft policy learning methods, where reverse KL divergence is used as the objective function. We assume that the Q-function is multimodal, and in combination with policy learning lagging behind, this leads to policy convergence towards a suboptimal distribution when minimizing reverse KL divergence.

As shown in Fig. 1(a), when the current policy is poorly initialized and far away from the optimal policy, exploring the optimal policy without a specific objective in mind can be challenging. This unguided exploration is inefficient and leads to poor performance, as the agent may fail to discover important states or actions necessary for achieving its objectives. The OAC algorithm attempts to combine Q-valued policy gradient information to guide exploration. It predicts an offset to allow better exploration, as shown in the figure; however, this offset cannot be accurately predicted in high-dimensional action spaces, both its value and direction.

**OOD exploration.** Therefore, it is beneficial to construct a policy that can guide exploration. Guided exploration is similar to performing a breadth-first policy search at a state, which can help address the issues associated with the policy-based approach.

A natural idea is to conduct out-of-distribution(OOD) exploration, which is more robust for policy learning compared to OOD optimization, as it is often associated with instability. However, the fundamental issue is that we still need some guidance, as random OOD exploration doesn't guarantee good results. Since we employ soft policy learning to train the policy, with Q-function learning leading the way, we can effectively use Q-values to guide excessive exploration. As depicted in Figure 1(b), Q-value-guided out-of-distribution (OOD) exploration is effective in preventing suboptimal policies and facilitates the sampling of actions with high Q-values.

## 4 IMPROVING EXPLORATION IN SOFT POLICY LEARNING

In this section, we will begin by presenting a novel Q-value update method. Following that, we will develop an effective exploration strategy and integrate the value update and action exploration based on a specific premise. Finally, we will illustrate how to meet this premise and learn an effective policy.

### 4.1 DYNAMIC DOUBLE Q SOFTMAX UPDATE

We introduce the Dynamic Double Q Softmax (DDQS) operator for updating Q-values. This operator is grounded in the double Q-function framework, where two distinct Q-functions are independently trained to estimate the value of state-action pairs. As described in the introduction, the definitions of the greedy Q-function and the conservative Q-function are as follows,

$$Q^{max}(s,a) = \max\{Q^1(s,a), Q^2(s,a)\}, Q^{min}(s,a) = \min\{Q^1(s,a), Q^2(s,a)\},$$

we denote $Q^{max}$ as the greedy Q-function and $Q^{min}$ as the conservative Q-function. The DDQS operator is defined as follows: for all $s$ in the state space $\mathcal{S}$,

$$ddqs_{\beta_t}(Q(s,\cdot)) = \frac{\sum_{a\in\mathcal{A}} e^{\beta_t Q^{max}(s,a)} Q^{min}(s,a)}{\sum_{a\in\mathcal{A}} e^{\beta_t Q^{max}(s,a)}}.$$

Here, $\beta_t$ represents a dynamically increasing hyper-parameter during the training iteration. We will now provide a theoretical analysis of the proposed DDQS operator and demonstrate that it offers a convergence guarantee.

A modified Bellman backup operator $\mathcal{T}^\pi$ given by

$$\mathcal{T}^\pi Q(\mathbf{s}_t, \mathbf{a}_t) \triangleq r(\mathbf{s}_t, \mathbf{a}_t) + \gamma \mathbb{E}_{\mathbf{s}_{t+1}\sim p}\left[V(\mathbf{s}_{t+1})\right],$$

where

$$V(\mathbf{s}_t) = ddqs_{\beta_t}(Q(\mathbf{s}_t,\cdot))$$

**Theorem 1** (Convergence of value iteration with the DDQS operator). *For any dynamic double Q softmax operator $ddqs_{\beta_t}$, if $\beta_t$ approaches $\infty$ after $t$ iterations, the value function $Q_t$ converges to the optimal value function $Q^*$.*

The proof is deferred to Appendix A.1. We expand upon the utilization of the DBS operator, as introduced in (Pan et al., 2020), within the double Q-function framework. This approach is less susceptible to overestimation. The motivation for this approach lies in the challenges posed by overestimation when learning the value function in continuous action spaces. Therefore, our goal is not solely to construct a policy $\pi_Q$, but rather to consider the combination of the two Q-functions to effectively address and mitigate overestimation issues.

## 4.2 Exploration with Greedy Q Value

Inspired by the results of the above theorems, we employ the double Q-function framework and utilized the greedy Q-value to construct a novel exploration policy, denoted as $\pi_E$. [1] We first define the exploration policy $\pi_E$,

$$\pi_E(\,\cdot\,|\mathbf{s}_t) = \frac{e^{\beta_t Q^{max}(\mathbf{s}_t,\,\cdot\,)}}{\sum_{a\in\mathcal{A}} e^{\beta_t Q^{max}(\mathbf{s}_t,a)}}.$$

Based on the results of Theorem 1, we can utilize the following formula to update the target Q-value:

$$r(\mathbf{s}_t,\mathbf{a}_t) + \gamma\,\mathbb{E}_{\mathbf{s}_{t+1}\sim p,\mathbf{a}_{t+1}\sim\pi_E(\,\cdot\,|\mathbf{s}_{t+1})}\left[Q(\mathbf{s}_{t+1},\mathbf{a}_{t+1})\right]. \tag{1}$$

Nevertheless, calculating the target values for the next state can be computationally intensive. It involves sampling across all possible states and actions and subsequently computing the corresponding Q-values. Drawing inspiration from the SARSA method, we can sample two consecutive (s, a) pairs to estimate the expectation of the Q-value:

$$\mathbb{E}_{\mathbf{s}_t\sim p,\mathbf{a}_t\sim\pi_E(\,\cdot\,|\mathbf{s}_t),\mathbf{s}_{t+1}\sim p,\mathbf{a}_{t+1}\sim\pi_E(\,\cdot\,|\mathbf{s}_{t+1})}\left[r(\mathbf{s}_t,\mathbf{a}_t) + \gamma Q(\mathbf{s}_{t+1},\mathbf{a}_{t+1})\right]. \tag{2}$$

The key distinction here is that we can estimate the expectation of the Q-value with finite sampling. The target Q-value (Eq. 1) necessitates evaluating the next Q-value across the entire state and action space. In contrast, consecutive pairs involve computation in an on-policy form. In continuous action RL tasks, we learn the policy separately from the Q-function. If we can sample actions from the policy $\pi$ as follows:

$$\mathbb{E}_{(\mathbf{s}_t,\mathbf{a}_t,\mathbf{s}_{t+1})\sim(p,\pi_E(\,\cdot\,|\mathbf{s}_{t+1}),p)}\left[r(\mathbf{s}_t,\mathbf{a}_t) + \gamma\,\mathbb{E}_{\mathbf{a}_{t+1}\sim\pi(\,\cdot\,|\mathbf{s}_{t+1})}\left[Q(\mathbf{s}_{t+1},\mathbf{a}_{t+1})\right]\right], \tag{3}$$

then, we can employ this equation to update the Q-function in an off-policy manner. It's evident that we have devised a new exploration strategy through this approach.

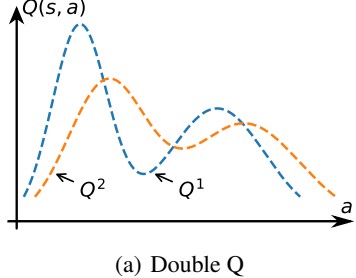
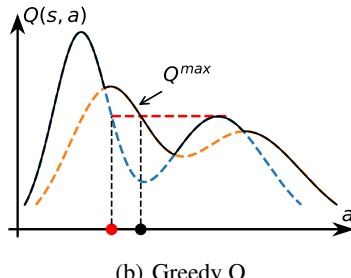

(a) Double Q               (b) Greedy Q

Figure 2: The state $s$ is fixed. Left: The two Q functions are in an energy-based form, which is the optimal solution for the maximum-entropy objective. Right: Greedy Q function take the maximum value of these two Q functions over the action space. The probability of reaching the red point increases significantly when we sample actions according to the value of the Greedy Q instead of $Q^1$. This strategy is more effective for escaping from sub-optimal states.

As depicted in Fig. 2, our proposed greedy Q exploration strategy offers several advantages: 1) This strategy proves superior for exploration compared to relying solely on any single Q-function or the

---

[1] $\pi_E$ is constructed using the greedy Q-value from the double Q-framework, while $\pi_Q$ is constructed using the Q-value from a single Q-network.

policy. As illustrated by the black and red points in the figure, the number of actions better than the suboptimal action increases, and their relative range expands. 2) The use of the max operator in our method is a form of overestimation. Overestimation can be problematic for Q-value updates, but it exhibits more favorable properties when employed for exploration purposes. 3) Although our method is named 'Bold,' it actually promotes exploration by diminishing the likelihood of selecting the action with the highest value. This is accomplished through overestimating the values of all available actions.

We also address the prerequisites for transitioning from Eq. 2 to Eq. 3. This transition necessitates that the actions sampled from the $\pi$ are as consistent as possible with the actions sampled from the $\pi_E$. In other words, ensuring that these two policies are as aligned as possible. Subsequently, we delve into the discussion of how to learn this policy $\pi$.

## 4.3 POLICY LEARNING

Drawing inspiration from soft policy learning methods, we begin by defining a conservative policy for optimization, which is defined as follows:

$$\pi_O(\,\cdot\,|\mathbf{s}_t) = \frac{e^{Q^{min}(\mathbf{s}_t,\,\cdot\,)}}{\sum_{a \in \mathcal{A}} e^{Q^{min}(\mathbf{s}_t,a)}},$$

then we can let the policy directly learn from the target policy like the soft policy learning as follows:

$$\pi' = \arg\min_{\pi} D_{\mathrm{KL}}\left(\pi(\,\cdot\,|\mathbf{s}_t)\,\|\,\pi_O(\,\cdot\,|\mathbf{s}_t)\right),$$

Now consider the neural network parameterized $Q_\theta$ function and policy $\pi_\phi$, thus,

$$Q^{max}(\mathbf{s}_t,\mathbf{a}_t) = \max\{Q_{\theta_1}(\mathbf{s}_t,\mathbf{a}_t), Q_{\theta_2}(\mathbf{s}_t,\mathbf{a}_t)\}, Q^{min}(\mathbf{s}_t,\mathbf{a}_t) = \min\{Q_{\theta_1}(\mathbf{s}_t,\mathbf{a}_t), Q_{\theta_2}(\mathbf{s}_t,\mathbf{a}_t)\} \tag{4}$$

Next, to minimize the expected KL-divergence policy objective,

$$
\begin{aligned}
J_\pi(\phi) &= \mathbb{E}_{\mathbf{s}_t \sim \mathcal{D}}\left[D_{\mathrm{KL}}\left(\pi_\phi(\,\cdot\,|\mathbf{s}_t)\,\|\,\pi_O(\,\cdot\,|\mathbf{s}_t)\right)\right] \\
&= \mathbb{E}_{\mathbf{s}_t \sim \mathcal{D}}\left[D_{\mathrm{KL}}\left(\pi_\phi(\,\cdot\,|\mathbf{s}_t)\,\|\,\exp(Q_\theta^{min}(\mathbf{s}_t,\,\cdot\,) - \log Z_\theta(\mathbf{s}_t))\right)\right] \\
&= \mathbb{E}_{\mathbf{s}_t \sim \mathcal{D}}\left[\mathbb{E}_{\mathbf{a}_t \sim \pi_\phi(\,\cdot\,|\mathbf{s}_t)}\left[\log \pi_\phi(\mathbf{a}_t|\mathbf{s}_t) - Q_\theta^{min}(\mathbf{s}_t,\mathbf{a}_t) + \log Z_\theta(\mathbf{s}_t)\right]\right],
\end{aligned}
\tag{5}
$$

where $Z_\theta(\mathbf{s}_t)$ is a constant for given state, $\mathcal{D}$ is a replay buffer and Eq. 5 requires sampling action from the policy $pi_\phi$. [2] To make the policy trainable, which means that policy parameters are differentiable, the action is reparameterized as follows:

$$\mathbf{a}_t = f_\phi(\epsilon_t;\mathbf{s}_t), \epsilon_t \sim \mathcal{N}(\mu, \sigma^2). \tag{6}$$

The gradient of $J_\pi(\phi)$ with respect to $\phi$ as follows:

$$
\begin{aligned}
\nabla_\phi J_\pi(\phi) &= \nabla_\phi \mathbb{E}_{\mathbf{s}_t \sim \mathcal{D}, \epsilon_t \sim \mathcal{N}}\left[\log \pi_\phi(\mathbf{a}_t|\mathbf{s}_t) - Q^{min}(\mathbf{s}_t,\mathbf{a}_t)\right] \\
&= \mathbb{E}_{\mathbf{s}_t \sim \mathcal{D}, \epsilon_t \sim \mathcal{N}}\left[\nabla_\phi \log \pi_\phi(\mathbf{a}_t|\mathbf{s}_t) - \nabla_\phi Q^{min}(\mathbf{s}_t,\mathbf{a}_t)|_{\mathbf{a}_t = f_\phi(\epsilon_t;\mathbf{s}_t)}\right],
\end{aligned}
\tag{7}
$$

since we utilize a neural network to parameterize both the policy and Q-function, we can employ a deep learning framework to perform the forward computations for the two terms in Eq. 7. The automatic gradient mechanism inherent to the framework will handle the backpropagation automatically. We can then derive an unbiased estimation of Eq. 7 using the following equation:

$$\hat{\nabla}_\phi J_\pi(\phi) = \nabla_\phi \log \pi_\phi(\mathbf{a}_t|\mathbf{s}_t) - \nabla_\phi Q^{min}(\mathbf{s}_t,\mathbf{a}_t)|_{\mathbf{a}_t = f_\phi(\epsilon_t;\mathbf{s}_t)}. \tag{8}$$

Here, refer to Eq. 3, we write the Q learning objective:

$$J_Q(\theta) = \mathbb{E}_{(\mathbf{s}_t,\mathbf{a}_t) \sim \mathcal{D}}\left[\frac{1}{2}(Q_\theta(\mathbf{s}_t,\mathbf{a}_t) - \hat{Q}(\mathbf{s}_t,\mathbf{a}_t))^2\right], \tag{9}$$

where $\hat{Q}(\mathbf{s}_t,\mathbf{a}_t) = r(\mathbf{s}_t,\mathbf{a}_t) + \gamma \mathbb{E}_{\epsilon_{t+1} \sim \mathcal{N}}\left[Q^{min}(\mathbf{s}_{t+1},\mathbf{a}_{t+1}) - \log \pi(\mathbf{a}_{t+1}|\mathbf{s}_{t+1})\right]$, and $\mathbf{a}_{t+1} = f_\phi(\epsilon_{t+1};\mathbf{s}_{t+1})$, the $-\log \pi(\mathbf{a}_{t+1}|\mathbf{s}_{t+1})$ term is due to the computation is based on the maximum entropy framework and the Q function is also in an energy-based form. The transition from replay buffer $\mathcal{D}$ is generated from the interaction of the policy $\pi_E$ and the environment. Then the gradient of the Q learning objective(Equation (9)) can be estimated with an unbiased estimator

$$\hat{\nabla}_\theta J_Q(\theta) = \nabla_\theta Q_\theta(\mathbf{a}_t,\mathbf{s}_t)\left(Q_\theta(\mathbf{s}_t,\mathbf{a}_t) - r(\mathbf{s}_t,\mathbf{a}_t) - \gamma Q^{min}(\mathbf{s}_{t+1},\mathbf{a}_{t+1}) + \gamma \log \pi(\mathbf{a}_{t+1}|\mathbf{s}_{t+1})\right).$$

---

[2] $\pi_\phi$ contains parameters that need to be learned, while $\pi_E$ and $\pi_O$ do not contain parameters.

### 4.4 The Bold Exploration Algorithm

In the Bold Actor Conservative Critic (BACC) algorithm (refer to Algorithm 1 in the appendix), several key steps are followed: 1)Dynamic Increase of $\beta_t$ (line 3): This is done to ensure the convergence of the Q-value, as described in section 4.1; 2)Action Sampling from Exploration Policy (line 5): Actions are sampled from the exploration policy to interact with the environment, as detailed in section 4.2; 3)Storing Transitions in Memory Buffer: The resulting transitions are stored in a memory buffer; 4)Updating the Q-function (line 12) and Actor (line 13): BACC samples transitions from the memory buffer to update both the Q-function and the actor, as explained in section 4.3.

In more detail, the policy network outputs both $\mu$ and $\sigma$ for Equation (6). We uniformly sample $s_n$ actions from the range $[\mu - s_r * \sigma, \mu + s_r * \sigma]$ to evaluate the sampled actions and construct the $\pi_E$ distribution. The hyperparameters $s_r$, $s_n$, and $\beta_t$ are three key parameters in our algorithm. Further details regarding the parameters and time cost will be discussed in Appendix E.

### 4.5 Related Work

**Exploration.** Classical exploration methods in reinforcement learning encompass $\epsilon$-greedy and UCB-1 (Auer et al., 2002). In policy gradient methods, policy-based exploration can be facilitated by leveraging information from the policy itself, such as entropy regularization. The deterministic policy gradient method (Silver et al., 2014) introduced the concept of separating policy learning from Q-function learning. Subsequently, deterministic-policy-based methods began exploring by randomly sampling actions around the policy, leading to the development of the Optimistic Actor-Critic (OAC) method (Ciosek et al., 2019). OAC predicts an offset of the Gaussian policy mean to encourage out-of-distribution (OOD) exploration. In the context of continuous RL tasks, several heuristic value-based exploration methods have emerged. For instance, the Coherent Exploration algorithm (Zhang & Van Hoof, 2021) directly modifies the last layer parameters of the policy network to enhance the policy's exploratory nature. The DOIE algorithm (Lobel et al., 2022) explores using a modified Q-function, assigning an optimistic value to transitions that lie significantly beyond the agent's prior experience. The RRS algorithm (Sun et al., 2022) directly alters Q-values, effectively adjusting the initialization parameters of the Q-network, but this requires prior knowledge, such as auxiliary rewards. In contrast, our method does not necessitate prior knowledge. In specific practical applications, the AW-OPT (Lu et al., 2022) algorithm uses both policy and Q value for exploration, assigning different weights to these two exploration approaches to achieve better control.

**Overestimation.** The concept of overestimation was first introduced in the paper by Thrun and Schwartz (Thrun & Schwartz, 1993), discussing the positive approximation error in the function approximation for RL. Then the MCQ-L (Rummery & Niranjan, 1994) method(famous with the name "SARSA" (Sutton & Barto, 2018)) mentioned that the argmax operator is impractical in training. They estimate the Q value with the consequent two-state-action pairs(in an online form). The Double Q learning (Hasselt, 2010) updates the Q value with two estimators to avoid overestimation. Then the Double DQN (Van Hasselt et al., 2016) is proposed, which parameterizes the Q function with a neural network. Inspired by the Double DQN, TD3 (Fujimoto et al., 2018) algorithm is proposed to handle overestimation for continuous action space. Regarding the stable Q-value updates, our method follows the TD3 approach, using the minimum Q-value from the two delayed-updated Q-functions to estimate the target Q-value, thereby minimizing overestimation. As for exploration, we harness overestimation to encourage bolder exploration strategies.

**Policy learning.** Stochastic policy gradient methods, such as A3C (Mnih et al., 2016), TRPO (Schulman et al., 2015), and PPO (Schulman et al., 2017), are applicable for policy learning in continuous action spaces. However, optimizing stochastic policy gradients in continuous action spaces can be challenging, and deterministic policy gradient methods based on value functions often yield better results. In the DPG (Silver et al., 2014) algorithm, the policy parameters are optimized to maximize the Q-function. The DDPG (Lillicrap et al., 2016) algorithm is a neural network-based variant of the DPG algorithm. The SQL (Haarnoja et al., 2017) algorithm assumes an energy-based form for the Q-function. The TD3 (Fujimoto et al., 2018) algorithm introduces conservative policy parameter updates using the minimum value of the two Q-functions. Lastly, the SAC (Haarnoja et al., 2018) algorithm employs a stochastic actor for policy exploration. Previous methods empirically calculate policy gradient with the minimum Q-value. In this paper, we explicitly define

the learning scheme, minimizing the KL-divergence between the target policy and the conservative policy.

## 5 EXPERIMENTS

We conducted experiments using the Mujoco physics engine (Todorov et al., 2012), which is currently freely available and maintained by DeepMind. Additionally, we utilized the PyBullet physics engine (Coumans & Bai, 2016–2021), which offers challenging environments for simulating robot control tasks. These simulations were interacted with through the Python API provided by OpenAI Gym (Brockman et al., 2016) for ease of use and interaction. In the following sections, we will present the primary experimental results along with their corresponding analyses. More specific and detailed results can be found in the appendix E. Unless otherwise specified, the units on the horizontal axis of the graph represent 1M steps.

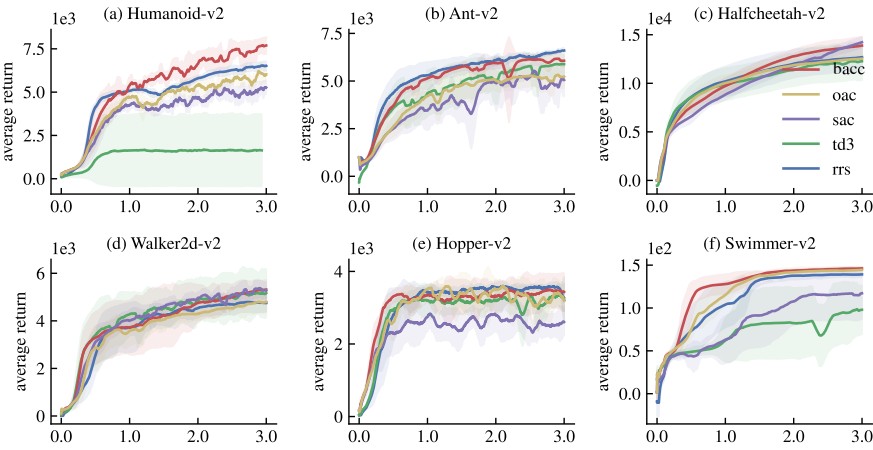

Figure 3: Results of BACC and four baseline algorithms in the six continuous environments

**General results on MuJoCo benchmark.** We compare BACC to OAC[2019] (Ciosek et al., 2019), SAC[2018] (Haarnoja et al., 2018), TD3[2018] (Fujimoto et al., 2018) and RRS[2022] (Sun et al., 2022) , four recent model-free RL methods that achieve state-of-the art performance. All methods run with six random seeds. The policy network and the Q network are the same for all methods. BACC uses three hyper-parameter related to exploration, which has been introduced in section 4.4. We provide the value of all hyper-parameter in the appendix C. The results are organized based on the complexity of the environment, ranging from complex to simple, as illustrated by Figure 3(a) through Figure 3(f). The Humanoid environment is the most complex, and the Swimmer environment is the simplest. The state dim and action dimension are summarized in the appendix D. As shown in Figure 3, our method achieves promising results on this benchmark. On Humanoid-v2, BACC achieves state-of-the-art performance and is sample efficient than previous algorithms. On Ant-v2, BACC works slightly worse than the RRS algorithm in the final performance. On Halfcheetah-v2, our method get better sample efficient. On Walker2d-v2 and Hopper-v2, our method get similar results with others. On Walker2d-v2, our method work better in the early learning stage.

**Assessment on exploration.** When evaluating the quality of exploration solely based on the results in Fig. 3, it might not provide a clear understanding of what is happening. Therefore, we conducted an analysis of the rewards obtained during each exploration. Comparing Fig. 3 and Fig. 4, we have the following observations: In the humanoid environment, the differences in exploration rewards are not very significant among the algorithms. However, there is a substantial difference in the learned policies. In the hopper environment, OAC's exploration seems to have become ineffective, but the algorithm continues to improve its policy. These observations reveal important insights: The first observation suggests that high-quality exploration has a significant impact on improving policy learning. The second observation indicates that off-policy algorithms are more robust to

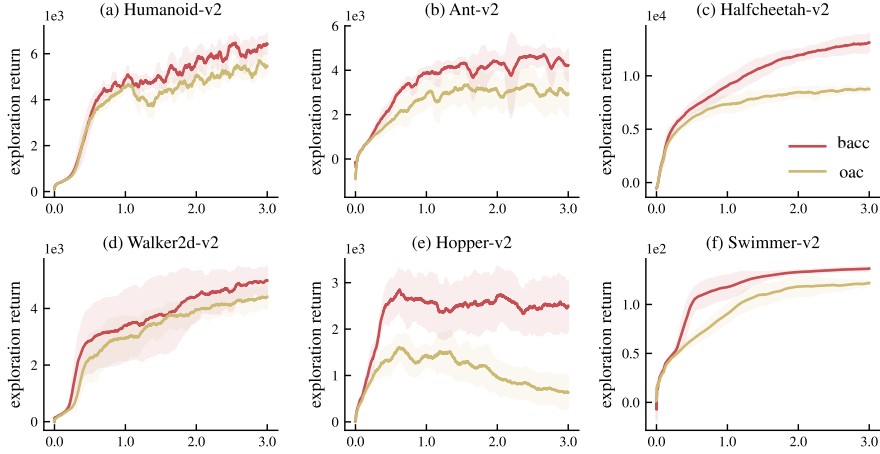

Figure 4: Average exploration episode return of BACC and OAC

suboptimal exploration results. In other words, poor exploration outcomes do not necessarily have a fatal impact on policy learning if the policy has been optimal (at about 0.8M steps). Additionally, this graph provides a more direct illustration of the effectiveness of our exploration strategy.

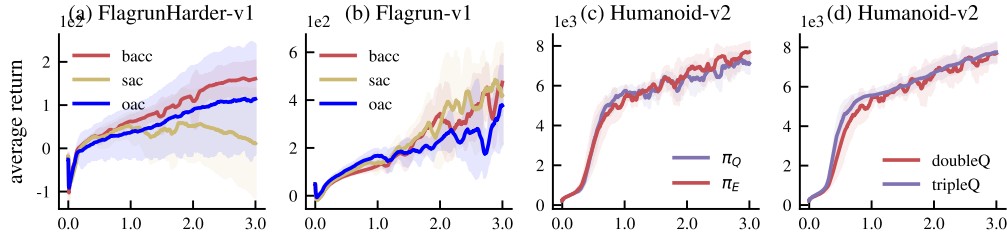

Figure 5: (a-b) More results on RoboSchool, Flagrun-v1 and FlagrunHarder-v1.(c) We compare the exploration result of $\pi_Q$ vs $\pi_E$ in the Humanoid-v2 environment. (d) We added additional Q-functions to see if the results would improve.

**Additional discussion.** We conducted experiments on the Roboschool simulation platform. We add comparative experiments in HumanoidFlagrun-v1 and HumanoidFlagrunHarder-v1 on Roboschool simulation. In the HumanoidFlagrun environment, the robot must run toward a randomly generated flag. In the HumanoidFlagrunHarder environment, the robot will be constantly bombarded by white cubes. In these two environments, the position of the flag is changed randomly, so the performances of the off-policy algorithm are pretty poor. In Fig. 5(a), it can be seen that using policy for exploration is better than using Q for exploration. In Fig. 5(b), since the environment is more difficult so that most of the interactions occur in the early stage, our method can be better when the information of Q can be properly utilized. In Fig. 5(c), we show that exploring with $\pi_E$ gives better results than $\pi_Q$, it indicates that our utilization of overestimation is effective. If the difference between the two Q-functions is larger, it should be more efficient. We are curious about the impact of adding more Q-functions on the results, so we tested the effect of three Q functions. In Fig. 5(d), it can be found that triple Q functions gives better and more stable results.

## 6 CONCLUSION

In this paper, we have developed a practical policy-based exploration strategy for deterministic policy reinforcement learning in continuous action spaces, realizing Q-value-guided out-of-distribution exploration. We conducted experiments on the Mujoco and RoboSchool benchmarks. In comparison to prior methods, our approach achieves more effective action exploration and demonstrates substantial improvements over previous approaches in the most complex Humanoid-v2 environments.

**Reproducibility statement.** We have included a detailed proof of the proposed theorem in Appendix A, a comprehensive algorithm description in Appendix B, and the experimental hyperparameters in Appendix C. Additionally, we provide our code in the supplementary material to facilitate the replication and verification of our results.

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

# Appendix

## A  PROOFS

### A.1  THEOREM 1

This proof (of Theorem 1) uses techniques from the proof of Theorem 1 in the paper (Pan et al., 2020), adapting them to the setting considered in this paper. An informal overview is as follows. The maximum of the two Q values is a particular case in the computation of the operator. Here, we mainly show this process also can guarantee convergence. We start with a proposition from the paper (Pan et al., 2020) that shows the relation between the soft-max and the log-sum-exp function.

**Proposition 1**

$$lse_\beta(\mathbf{X}) - sm_\beta(\mathbf{X}) = \frac{1}{\beta}H(\mathbf{X}) = \frac{1}{\beta}\sum_{i=1}^{n} -p_i \log(p_i) \leq \frac{\log(n)}{\beta}, \tag{10}$$

where $p_i = \frac{e^{\beta x_i}}{\sum_{j=1}^{n} e^{\beta x_j}}$ denotes the weights of the softmax distribution, $lse_\beta(\mathbf{X})$ denotes the log-sum-exp function $lse_\beta(\mathbf{X}) = \frac{1}{\beta}\log(\sum_{i=1}^{n} e^{\beta x_i})$, and $sm_\beta(\mathbf{X})$ denotes the softmax function $sm_\beta(\mathbf{X}) = \frac{\sum_{i=1}^{n} e^{\beta x_i} x_i}{\sum_{j=1}^{n} e^{\beta x_j}}$. $H(\mathbf{X})$ is the entropy of the distribution. It is easy to check that the maximum entropy is achieved when $p_i = \frac{1}{n}$, where the entropy equals to $\log(n)$.

*Proof.*

$$\frac{1}{\beta}\sum_{i=1}^{n} -p_i \log(p_i)$$

$$= \frac{1}{\beta}\sum_{i=1}^{n}\left(-\frac{e^{\beta x_i}}{\sum_{j=1}^{n} e^{\beta x_j}}\log\left(\frac{e^{\beta x_i}}{\sum_{j=1}^{n} e^{\beta x_j}}\right)\right)$$

$$= \frac{1}{\beta}\sum_{i=1}^{n}\left(-\frac{e^{\beta x_i}}{\sum_{j=1}^{n} e^{\beta x_j}}\left(\beta x_i - \log\left(\sum_{j=1}^{n} e^{\beta x_j}\right)\right)\right)$$

$$= -\sum_{i=1}^{n}\frac{e^{\beta x_i} x_i}{\sum_{j=1}^{n} e^{\beta x_j}} + \frac{1}{\beta}\log\left(\sum_{j=1}^{n} e^{\beta x_j}\right)\frac{\sum_{i=1}^{n} e^{\beta x_i}}{\sum_{j=1}^{n} e^{\beta x_j}}$$

$$= -sm_\beta(\mathbf{X}) + lse_\beta(\mathbf{X})$$

$\square$

**Theorem 1 (Convergence of value iteration with the DDQS operator)**  *For any dynamic double Q softmax operator $gdq_{\beta_t}$, if $\beta_t$ approaches $\infty$, the value function after $t$ iterations $v_t$ converges to the optimal value function $V*$.*

*Proof.*

$$||(\mathcal{T}_{\beta_t} V_i) - (\mathcal{T}_{\beta_t} V_j)||_\infty \tag{11}$$

$$= \max_s |ddqs_{\beta_t}(Q_i^{max}(s,\cdot)) - ddqs_{\beta_t}(Q_j(s,\cdot))| \tag{12}$$

$$\leq \max_s |lse_{\beta_t}(Q_i(s,\cdot)) - \frac{1}{\beta}H(Q_i(s,\cdot))$$

$$- lse_{\beta_t} Q_j(s,\cdot)) + \frac{1}{\beta}H(Q_j(s,\cdot))| \tag{13}$$

$$\leq \underbrace{\max_s |lse_{\beta_t}(Q_i(s,\cdot)) - lse_{\beta_t}((Q_j(s,\cdot))|}_{(A)} + \underbrace{\frac{\log(|A|)}{\beta_t}}_{(B)}, \tag{14}$$

For the term $(A)$, the log-sum-exp operator has been proved as a non-expanding operator in the paper (Fox et al., 2015). In mathematics, the log-sum-exp function is almost equal to the max function. That is why this makes sense. Here, we prove that it's also a non-expanding operator in our greedy Q setting. Define a norm on Q value as $\|Q_i - Q_j\| \triangleq \max_{\mathbf{s,a}} |Q_i(\mathbf{s,a}) - Q_j(\mathbf{s,a})|$. Suppose $\epsilon = \|Q_i - Q_j\|$. Please note that in our setting, $Q_i(\mathbf{s,a}) = \min Q_i^1(\mathbf{s,a}), Q_i^2(\mathbf{s,a})$, $Q_j(\mathbf{s,a})$ is similar defined. Then

$$\log \int \exp(Q_i(\mathbf{s',a'}))\, d\mathbf{a'} \leq \log \int \exp(Q_j(\mathbf{s',a'}) + \epsilon)\, d\mathbf{a'}$$
$$= \log \left( \exp(\epsilon) \int \exp Q_j(\mathbf{s',a'})\, d\mathbf{a'} \right)$$
$$= \epsilon + \log \int \exp Q_j(\mathbf{a',a'})\, d\mathbf{a'}. \tag{15}$$

Similarly, $\log \int \exp Q_i(\mathbf{s',a'})\, d\mathbf{a'} \geq -\epsilon + \log \int \exp Q_j(\mathbf{s',a'})\, d\mathbf{a'}$. Therefore $\|\mathcal{T}Q_i - \mathcal{T}Q_j\| \leq \gamma\epsilon = \gamma\|Q_i - Q_j\|$.

Consider for $Q^1$ and $Q^2$, we expand the composed Q function, we derive

$$\|\mathcal{T}Q_i - \mathcal{T}Q_j\| \leq \gamma\|Q_i - Q_j^1\|, \|\mathcal{T}Q_i - \mathcal{T}Q_j\| \leq \gamma\|Q_i - Q_j^2\|$$

when $Q_i = Q_i^1$, it means $Q_i^1$ is the min value, for the term $(A)$, we have

$$\max_s |\text{lse}_{\beta_t}(Q_i(s,\cdot)) - \text{lse}_{\beta_t}((Q_j(s,\cdot))|$$
$$\leq \max_{s,a} \frac{1}{\beta_t}|\beta_t Q_i^1 - \beta_t Q_j^1|$$
$$\leq \gamma \max_{s,a} \sum_{s'} p(s'|s,a)|V_i(s') - V_j(s')|$$
$$\leq \gamma \|V_i - V_j\|_\infty$$

So $\mathcal{T}$ is a contraction. We can get the same result when $Q_i = Q_i^2$. And due to the min operator, $Q^1$ and $Q^2$ are updated iteratively. Consequently, the two Q functions converge to the optimal value, satisfying the modified Bellman equation. Thus, the optimal policy is unique.

For the term $(B)$, the details can be found in (Pan et al., 2020). A direct understanding is that as $\beta$ increases, this term will eventually become zero. □

## B Algorithm

Compared to SAC and OAC algorithms, the BACC algorithm 1 primarily makes modifications in the exploration component. In practical applications, instead of calculating the state-value function, BACC directly updates the Q-functions. In our algorithm, several key steps are followed: 1)Dynamic Increase of $\beta_t$ (line 3): This is done to ensure the convergence of the Q-value, as described in section 4.1; 2)Action Sampling from Exploration Policy (line 5): Actions are sampled from the exploration policy to interact with the environment, as detailed in section 4.2; 3)Storing Transitions in Memory Buffer: The resulting transitions are stored in a memory buffer; 4)Updating the Q-function (line 12) and Actor (line 13): BACC samples transitions from the memory buffer to update both the Q-function and the actor, as explained in section 4.3.

## C Hyper parameters

Table 1 provides an overview of the common BACC parameters utilized in the comparative evaluation presented in Figure 3. However, for the results in Figure 5(a-b), which were evaluated on the RoboSchool benchmark, we made slight adjustments to the hyperparameters. In the HumanoidFlagrunHarder-v1 environment, we set $\beta_t$ to 10, and $s_r$ to 3. In the HumanoidFlagrun-v1 environment, we used $\beta_t$=1 and $s_r$=0.1.

---

**Algorithm 1** Bold Actor and Conservative Critic (BACC).

---

**Require:** $\theta_1, \theta_2, \phi$      ▷ Initial parameters $\theta_1, \theta_2$ of the Q function and $\phi$ of the target policy $\pi_T$.

1:   $\breve{\theta}_1 \leftarrow \theta_1, \breve{\theta}_2 \leftarrow \theta_2, \mathcal{D} \leftarrow \emptyset$      ▷ Initialize target network weights and replay buffer
2:   **for** each iteration **do**
3:      increase $\beta_t$ according to the iteration number
4:      **for** each environment step **do**
5:         $\mathbf{a}_t \sim \pi_E(\mathbf{a}_t|\mathbf{s}_t, \beta_t)$      ▷ Sample action from exploration policy as in (4.2).
6:         $\mathbf{s}_{t+1} \sim p(\mathbf{s}_{t+1}|\mathbf{s}_t, \mathbf{a}_t)$      ▷ Sample state from the environment
7:         $\mathcal{D} \leftarrow \mathcal{D} \cup \{(\mathbf{s}_t, \mathbf{a}_t, R(\mathbf{s}_t, \mathbf{a}_t), \mathbf{s}_{t+1})\}$      ▷ Store the transition in the replay buffer
8:      **end for**
9:      **for** each training step **do**
10:         sample batch transition $(\mathbf{s}_t, \mathbf{a}_t, R(\mathbf{s}_t, \mathbf{a}_t), \mathbf{s}_{t+1})$ from the buffer
11:         compute target $\hat{Q}^i(\mathbf{s}_t, \mathbf{a}_t)$ for $i \in 1, 2$
12:         update $\theta_i$ with $\hat{\nabla}_{\theta_i} J_Q(\theta_i)$ for $i \in 1, 2$      ▷ Q parameter update
13:         update $\phi$ with $\hat{\nabla}_{\phi} J_{\pi}(\phi)$      ▷ Policy parameter update
14:         $\breve{\theta}_1 \leftarrow \tau\theta_1 + (1-\tau)\breve{\theta}_1, \breve{\theta}_2 \leftarrow \tau\theta_2 + (1-\tau)\breve{\theta}_2$      ▷ Update target networks
15:      **end for**
16: **end for**

**Ensure:** $\theta_1, \theta_2, \phi$      ▷ Optimized parameters

---

| Parameter | Value |
|---|---|
| *Shared* | |
|     optimizer | Adam |
|     learning rate | $3 \cdot 10^{-4}$ |
|     discount ($\gamma$) | 0.99 |
|     replay buffer size | $10^6$ |
|     number of hidden layers (all networks) | 2 |
|     number of hidden units per layer | 256 |
|     number of samples per minibatch | 256 |
|     nonlinearity | ReLU |
| *SAC* | |
|     target smoothing coefficient ($\tau$) | 0.005 |
|     target update interval | 1 |
|     gradient steps | 1 |
| *OAC* | |
|     beta UB | 4.66 |
|     delta | 23.53 |
| *BACC* | |
|     dynamic weight ($\beta_t$) | epoch number*1 |
|     sample range ($s_r$) | 7 |
|     sample size ($s_n$) | 32 |

Table 1: BACC Hyper-parameters

## D   ENVIRONMENT PROPERTIES

The properties of each environment are summarized in Table 2.

| Environment | State dim | Action dim | Episode Length |
|---|---|---|---|
| Humanoid-v2 | 376 | 17 | 1000 |
| Ant-v2 | 111 | 8 | 1000 |
| HalfCheetah-v2 | 17 | 6 | 1000 |
| Walker2d-v2 | 17 | 6 | 1000 |
| Hopper-v2 | 11 | 3 | 1000 |
| Swimmer-v2 | 8 | 2 | 1000 |

Table 2: The details of Mujoco Environments used in this paper.

## E    MORE RESULTS

### E.1    RELATION BETWEEN BETA AND SAMPLE SIZE

We sample $n$ integers uniformly from [1,1000] and give the numerical result of the $\text{lse}_\beta(\mathbf{X})$, $\text{sm}_\beta(\mathbf{X})$ and $\frac{1}{\beta}H(\mathbf{X})$. According to the results shown in Table 3,Table 4,Table 5, we can see that a large beta will lead $\text{sm}_\beta(\mathbf{X})$ give a consistent result with the max operator. And, $\text{sm}_\beta(\mathbf{X})$ is better than $\text{lse}_\beta(\mathbf{X})$, it approximate the maximum from the lower bound. With a large beta, the maximum value can be aprroximated regardless of the sample size.

| $n$, $beta$=0.01 | $\text{lse}_\beta(\mathbf{X})$ | $\text{sm}_\beta(\mathbf{X})$ | $\frac{1}{\beta}H(\mathbf{X})$ | maximum |
|---|---|---|---|---|
| 10 | 992.46 | 932.61 | 59.85 | 976 |
| 100 | 1252.17 | 900.62 | 351.50 | 995 |
| 1000 | 1461.84 | 899.60 | 561.094 | 999 |
| 10000 | 1692.55 | 900.98 | 779.09 | 999 |
| 100000 | 1920.21 | 899.25 | 896.54 | 999 |
| 1000000 | 2150.74 | 899.56 | nan | 999 |

Table 3: The results when $beta$=0.01.

| $n$, $beta$=1 | $\text{lse}_\beta(\mathbf{X})$ | $\text{sm}_\beta(\mathbf{X})$ | $\frac{1}{\beta}H(\mathbf{X})$ | maximum |
|---|---|---|---|---|
| 10 | 834.69 | 834.00 | 0.69 | 834 |
| 100 | 970.02 | 969.92 | 0.09 | 970 |
| 1000 | 1000.29 | 998.70 | 1.58 | 999 |
| 10000 | 1001.69 | 998.31 | 3.38 | 999 |
| 100000 | 1004.10 | 998.41 | 5.68 | 999 |
| 1000000 | 1006.36 | 998.41 | 7.85 | 999 |

Table 4: The results when $beta$=1.

### E.2    REWARD DIFFERENCE BETWEEN EXPLORATION AND EVALUATION

As shown in Figure 6, Because both algorithms are related to exploration, the evaluation return is higher than the exploration return. However, something goes wrong in OAC exploration. As we use transition sampled from the replay buffer to train the policy, it does not seem to have much impact on policy learning. Instead, it shows that our exploration strategy is better than the OAC method.

### E.3    REWARD COMPARISON BETWEEN BACC AND OAC

As shown in Figure 7, our method performs better in policy evaluation. Another thing to note is that our method is inspired by OAC, and we found some problems with the exploration strategy of OAC. Therefore, we need to prove that our exploration strategy is better, as shown in Figure 8, this figure shows that our exploration strategy is better.

| $n, beta$=100 | $\mathrm{lse}_\beta(\mathbf{X})$ | $\mathrm{sm}_\beta(\mathbf{X})$ | $\frac{1}{\beta}H(\mathbf{X})$ | maximum |
|---|---|---|---|---|
| 10 | 947.0 | 947.0 | 0.0 | 947 |
| 100 | 996.0 | 996.0 | 0.0 | 996 |
| 1000 | 997.0 | 997.0 | 0.0 | 997 |
| 10000 | 999.02 | 999.00 | 0.02 | 999 |
| 100000 | 999.05 | 999.0 | 0.05 | 999 |
| 1000000 | 999.07 | 999.0 | 0.07 | 999 |

Table 5: The results when $beta$=100.

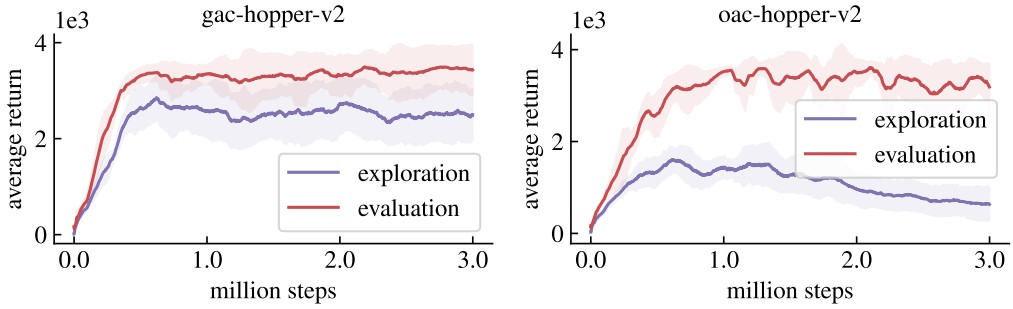

Figure 6: Exploration and evaluation difference of BACC and OAC

### E.4 MORE EXPERIMENTS

**Learning the Q value.** The parameter $\beta_t$ influences the learning of the Q function. In practice, we dynamically increase the $\beta_t$ by setting it as the multiplication of $\beta$ and the epoch number of timestep $t$. In the early stage of training, the Q function cannot provide little information, so a small $\beta$ can be used to encourage the exploration. The different $\beta$ results are shown in Figure 9(a). As we can see, a smaller $\beta$ can produce a slightly better result. Our method is not so sensitive to this parameter in our setting. Nevertheless, a smaller $\beta$ indeed takes a better result.

**Better exploration.** As shown in Figure 9(b), if we use a smaller value of $s_r$, that is, we sample action around the current policy, we can see the final result compared to the two larger values is terrible, which shows that **aimless exploration** does lead to poor results, it is difficult to get good results if $s_r$ is too small. Policy learning depends on how well the initial policy is. When $s_r$=7, we get better results indicating that the **policy divergence** problem can be solved by explicit sampling point out-of-distribution. Thus, we can do more OOD sampling with the Q function to explore action space. Additionally, according to the results of $s_r$=7 and $s_r$=9, we know that $s_r$ should not be as big as possible. Bigger $s_r$ does not mean better results because the policy gradient comprises the $\nabla_\phi Q$ and $\nabla_\phi \pi$. Suppose we optimize the policy with many low-probability actions, which may prefer by the Q function. In that case, the policy gradient may be too small to promote the parameter update.

**Exploration policy.** When constructing an exploration policy, a key question is how many actions need to be evaluated to obtain a usable exploration policy. If constructing the policy needs many actions to be evaluated, our method becomes computationally burdensome and resource-intensive. As shown in Figure 9(c), when $s_n$=10, the final result is the best; when $s_n$=1000, the best result is achieved at about one million steps, but the final result does not gain an advantage. According to the results in the figure, it can be seen that BACC can be effective by evaluating only a limited number of actions, and the improvement of the final performance does not lie in the complete evaluation of actions in the action space. Instead, the exploration strategy is the main factor for performance improvement.

**Time cost.** Our experimental setup consists of a computer with Ubuntu 18 operating system, equipped with a 9900K CPU and an RTX 2060 GPU. Without using the exploration strategy pro-

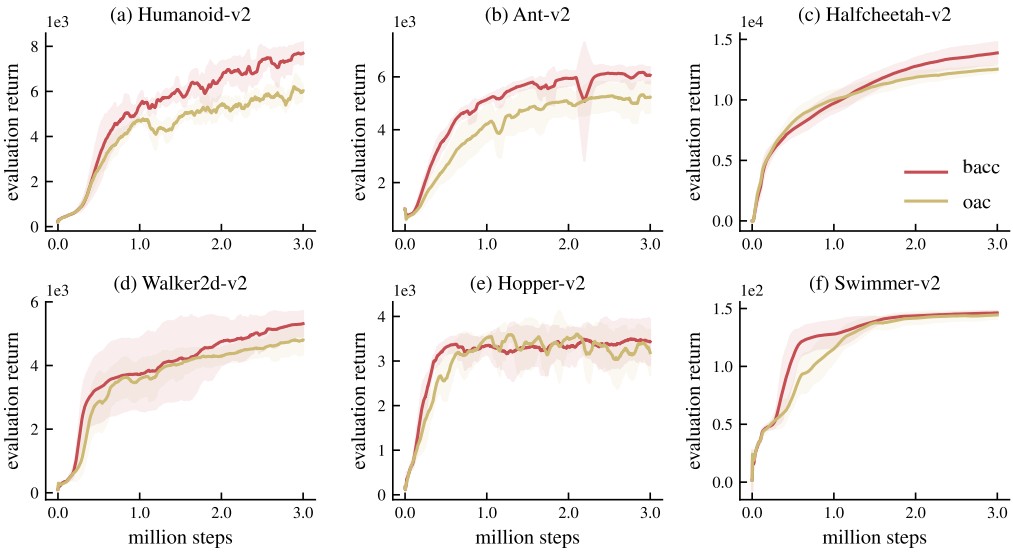

Figure 7: Average evaluation episode return of BACC and OAC

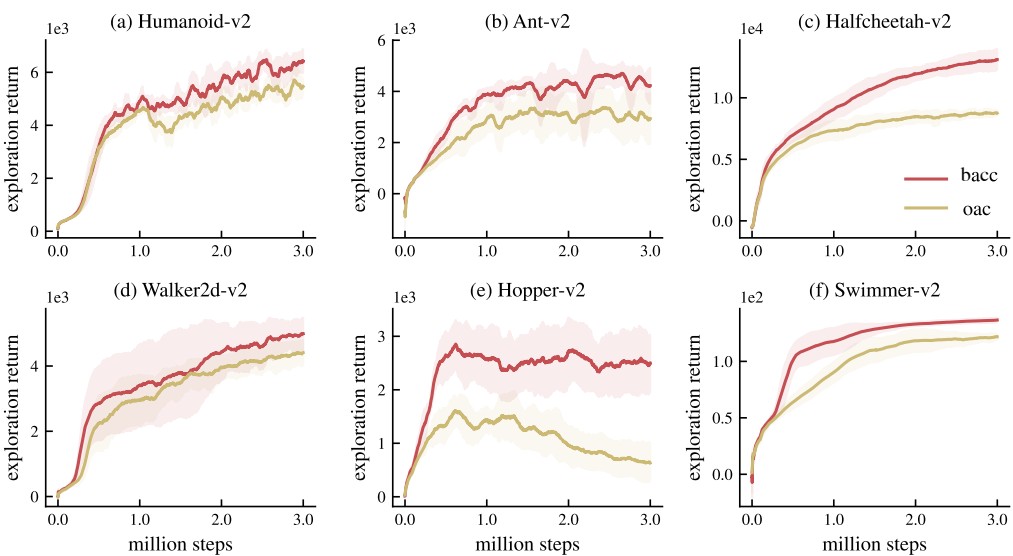

Figure 8: Average exploration episode return of BACC and OAC

posed in this paper, it takes 6.619 seconds to complete one epoch on average. In most of our experiments, we evaluate 32 actions. With this setting, running one epoch takes an average of 7.29 seconds. If we evaluate 64 actions, running one epoch takes an average of 7.09 seconds. The GPU may be more efficient when computing data with a batch size 64. From this, the additional time added due to exploration is insignificant. One epoch requires 1000 interactions with the environment. When averaged per exploration step, the time consumed is almost negligible.

We show all the results of different hyper-parameters on these six environments. As shown in Figure 10, Figure 11 and Figure 12, combine the numerical results, we can better choose the value for hyper-parameters.

### E.5   VISUALIZATION FOR THE Q VALUE

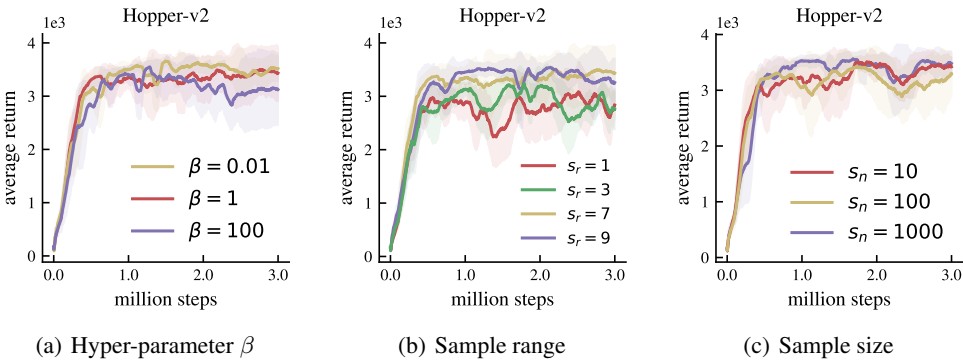

Figure 9: Three hyper-parameters related to the exploration policy. (a): In practice, the value of $\beta_t$ is obtained by multiplying $\beta$ with the epoch number of timestep $t$. (b): The parameter $s_r$ determines the sample range, where a large value indicates that sampled actions could deviate further from the distribution of the current policy. (c): We uniformly sample $s_n$ actions within the sample range to construct our exploration policy.

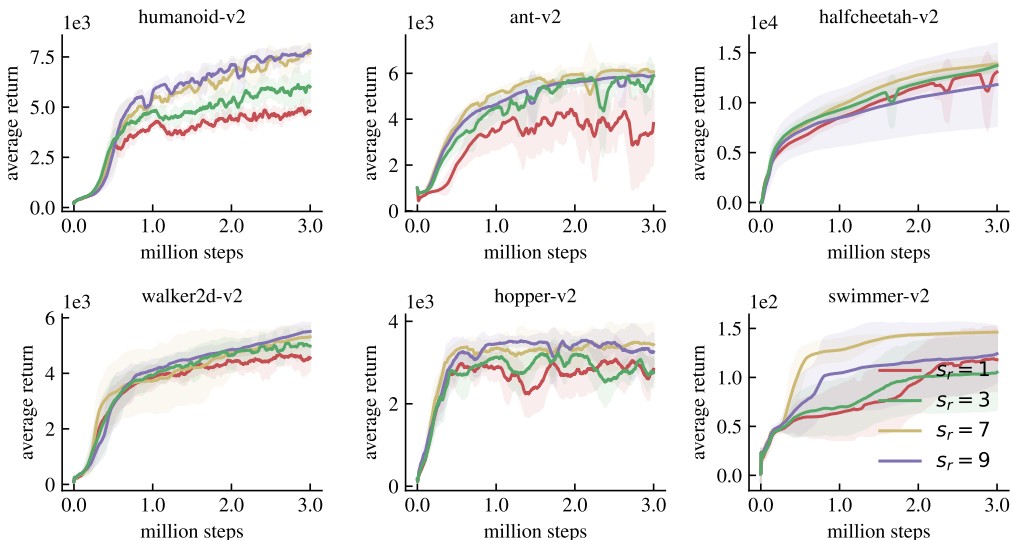

Figure 10: The results of different sample range on the six environments.

**Visualization of the Q function.** Our approach is based on the maximum entropy framework and assumes that the Q function is in an energy-based form. We design experiments to validate this assumption. The action space in the Simmer environment is two-dimensional, making it an ideal validation environment. We select an intermediate state of the Q network during the training process, sample 400*400 points across the entire action space, and calculate the corresponding Q values. The results we obtained are shown in Figure 13. We plot the 3d surface of the Q function, and a 2d plane for rotor2=-1.

To observe the surface of the Q network, we plot different stage Q value, which is evaluated based on a random start state and sampled actions in the Swimmer-v2 environments. As shown in Figure 14,in the initial stage, the surface is not flat, which is influenced by the input (state and action); if the state and action is zero vector, this surface should be flat, all zero. This phenomenon shows that neural networks imply prior knowledge about choosing actions. Policy initialization is closely related to policy learning. As training progresses, the final optimal action dramatically differs from the initial policy.

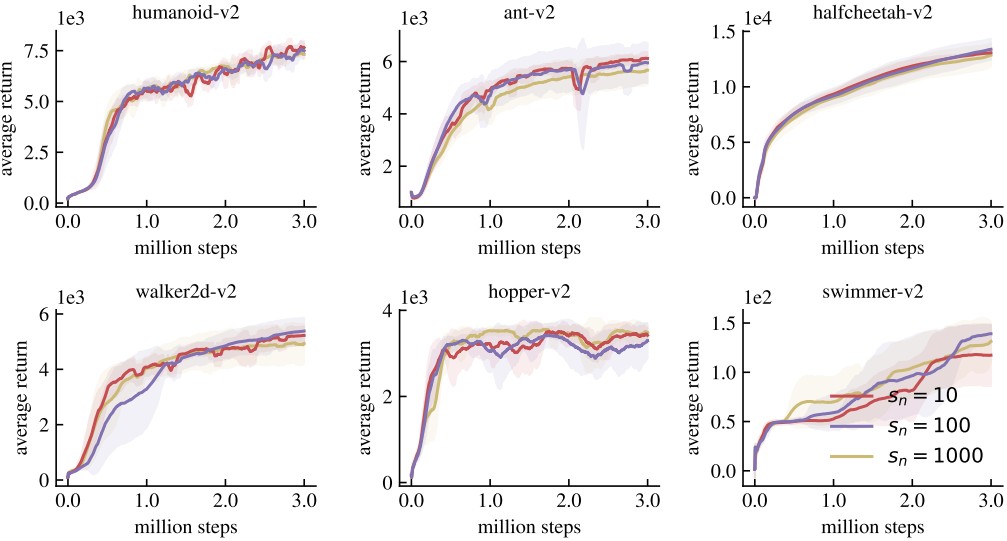

Figure 11: The results of different sample size on the six environments.

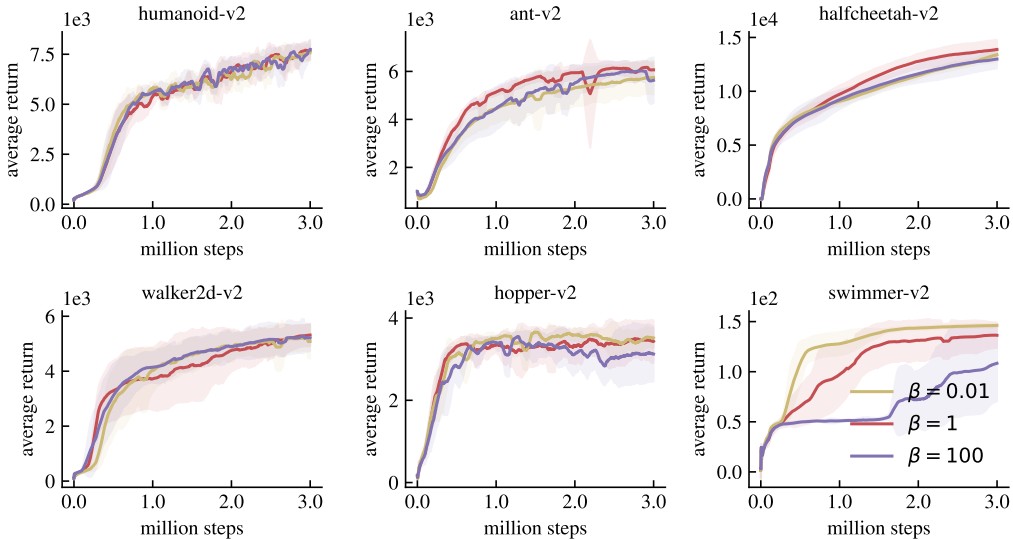

Figure 12: The results of different beta on the six environments.

## F   NUMERICAL RESULTS

Table F lists the numerical results for three time steps: 1e6, 2e6, and 3e6 timesteps. In different environments, the optimal results for these three time steps are all displayed in bold.

## G   LIMITATIONS AND BROADER IMPACTS

**Limitations.**   In low-dimensional action spaces, our method shows little improvement. It is particularly noticeable that in the swimmer-v2 environment, state-of-the-art results can reach an episode reward of 350. Furthermore, exploration costs should also increase as the action space's dimensionality increases. The conclusions we drew earlier may have limitations. However, it is challenging to

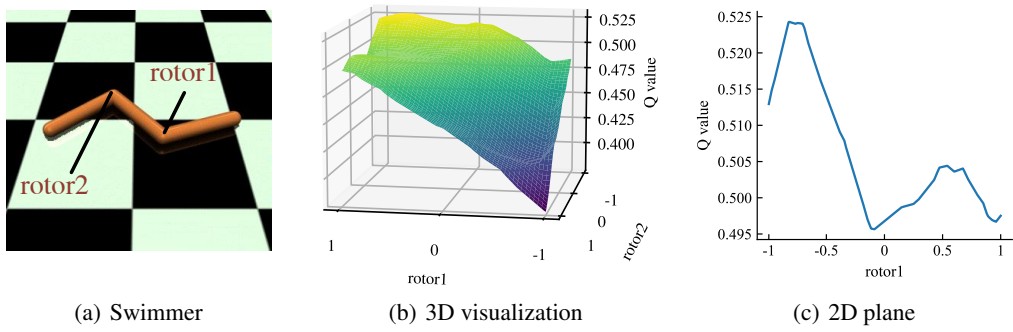

(a) Swimmer     (b) 3D visualization     (c) 2D plane

Figure 13: Visualization of the Q function. (a): The swimmer has two rotors, and its moving is controlled by adjusting the torque applied to the two rotors. (b): The Q values of the two-dimension actions are plotted in 3D Space. (c): We plot a particular case for rotor2=-1 to show that the Q function has an energy-based form in early-stage training.

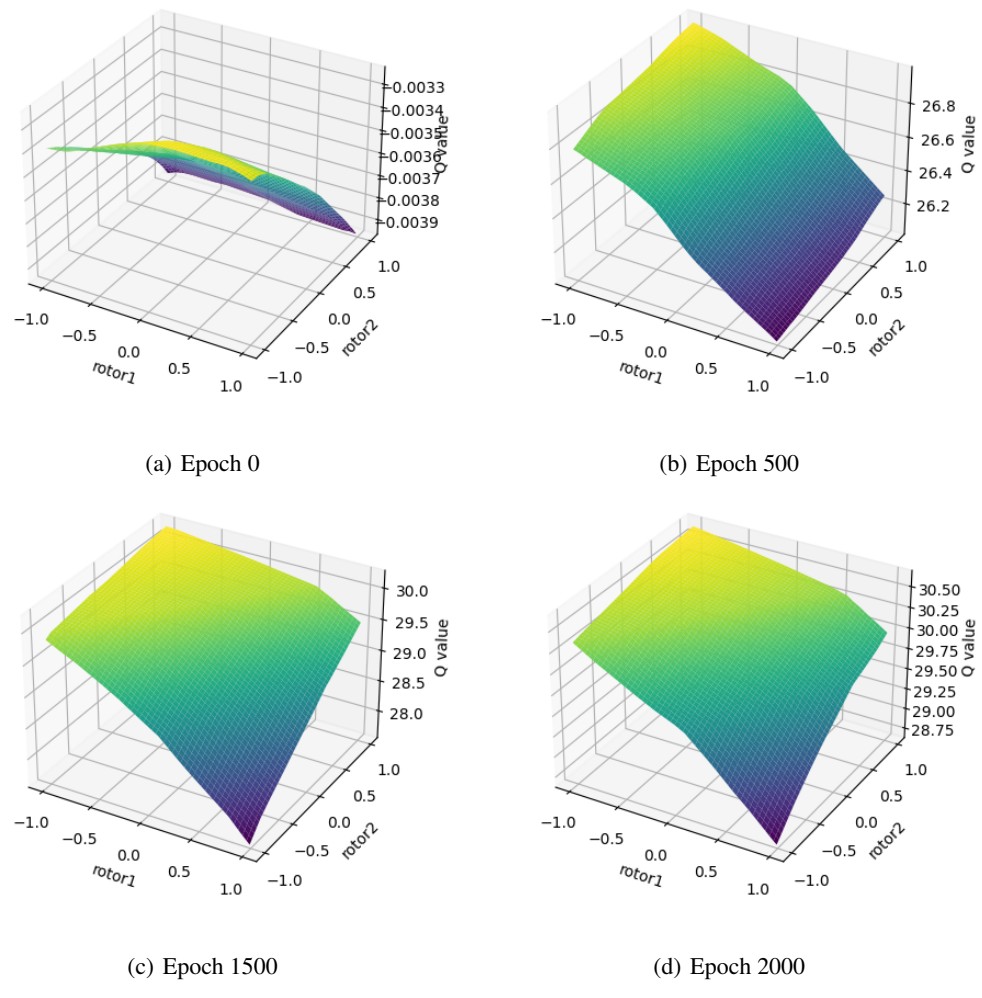

(a) Epoch 0     (b) Epoch 500

(c) Epoch 1500     (d) Epoch 2000

Figure 14: 3D surfaces of different epoch Q function

| Humanoid-v2($\mu \pm \sigma$) | 1e6 | 2e6 | 3e6 |
|---|---|---|---|
| td3 | 1756.27 ± 2238.27 | 1484.93 ± 1900.97 | 1598.9 ± 2036.04 |
| sac | 4567.02 ± 1685.49 | 5261.63 ± 963.99 | 4502.22 ± 2095.16 |
| oac | 4576.88 ± 814.27 | 5227.4 ± 1836.92 | 5819.47 ± 1933.58 |
| rrs | 5023.11 ± 372.79 | 5944.82 ± 51.67 | 6887.61 ± 186.2 |
| bacc | **5839.09 ± 444.96** | **7040.95 ± 460.54** | **7911.13 ± 616.55** |

| Ant-v2($\mu \pm \sigma$) | 1e6 | 2e6 | 3e6 |
|---|---|---|---|
| td3 | 4476.55 ± 1202.99 | 5566.91 ± 807.15 | 6019.93 ± 501.09 |
| sac | 3650.65 ± 1223.69 | 4883.99 ± 2146.68 | 5226.24 ± 894.48 |
| oac | 4298.93 ± 573.6 | 5037.22 ± 535.36 | 5495.85 ± 615.79 |
| rrs | **5452.52 ± 778.4**1 | **6527.35 ± 182.4** | **6588.18 ± 322.39** |
| bacc | 5080.74 ± 564.6 | 5990.94 ± 327.87 | 6142.29 ± 466.41 |

| Halfcheetah-v2($\mu \pm \sigma$) | 1e6 | 2e6 | 3e6 |
|---|---|---|---|
| td3 | 9884.53 ± 1649.81 | 11313.22 ± 1688.05 | 12495.93 ± 2317.8 |
| sac | 9020.98 ± 514.63 | 11961.11 ± 310.73 | **14339.73 ± 562.87** |
| oac | 10057.67 ± 880.59 | 11958.49 ± 591.72 | 12653.99 ± 333.77 |
| rrs | **10196.44 ± 1596.95** | 12186.35 ± 1456.34 | 12782.87 ± 1339.96 |
| bacc | 9915.54 ± 855.0 | **12985.18 ± 767.3** | 14021.4 ± 895.09 |

| Walker2d-v2($\mu \pm \sigma$) | 1e6 | 2e6 | 3e6 |
|---|---|---|---|
| td3 | **4431.91 ± 912.77** | **4957.68 ± 761.62** | **5447.56 ± 992.39** |
| sac | 4088.68 ± 590.64 | 4656.63 ± 819.8 | 5261.41 ± 389.63 |
| oac | 3582.6 ± 584.79 | 4024.35 ± 642.66 | 4919.95 ± 408.12 |
| rrs | 3771.38 ± 1128.64 | 4517.25 ± 344.15 | 4723.65 ± 424.02 |
| bacc | 3775.62 ± 1247.12 | 4846.29 ± 638.8 | 5324.82 ± 464.94 |

| Hopper-v2($\mu \pm \sigma$) | 1e6 | 2e6 | 3e6 |
|---|---|---|---|
| td3 | 3182.54 ± 594.2 | 2463.6 ± 1337.51 | 3409.52 ± 306.04 |
| sac | 2820.05 ± 730.76 | 2494.56 ± 910.32 | 2464.37 ± 1238.7 |
| oac | **3548.57 ± 95.57** | 3373.07 ± 512.82 | 2833.11 ± 1160.81 |
| rrs | 3469.71 ± 263.87 | **3451.62 ± 555.41** | 3125.02 ± 1259.31 |
| bacc | 3392.43 ± 230.12 | 3072.41 ± 655.76 | **3591.63 ± 342.43** |

| Swimmer-v2($\mu \pm \sigma$) | 1e6 | 2e6 | 3e6 |
|---|---|---|---|
| td3 | 66.29 ± 30.45 | 81.7 ± 42.95 | 96.14 ± 31.91 |
| sac | 63.34 ± 25.78 | 95.73 ± 40.15 | 118.05 ± 31.69 |
| oac | 118.62 ± 8.17 | 143.22 ± 6.4 | 145.19 ± 3.76 |
| rrs | 103.89 ± 26.15 | 137.55 ± 4.76 | 138.46 ± 3.43 |
| bacc | **128.39 ± 11.35** | **144.46 ± 1.87** | **147.78 ± 1.58** |

develop environments with higher-dimensional action spaces, and we still need to fully validate our conclusions in such environments.

**Broader impacts.** We do not anticipate any negative consequences from using our method in practice.

