# OpenReview forum: "Careful at Estimation and Bold at Exploration for Deterministic Policy Gradient Algorithm"
_ICLR.cc/2024/Conference — Submitted to ICLR 2024_

### Official Review · Reviewer_vPPz · 2023-10-14

**Soundness:** 3 good
**Presentation:** 3 good
**Contribution:** 3 good
**Rating:** 6
**Confidence:** 4

**Summary:**

This paper proposes a novel approach called Bold Actor Conservative Critic (BACC) to address the problems of unguided exploration and exclusive policy in policy learning. BACC uses Q-value to guide exploration. Suppose the high Q-value actions far from the policy can also be sampled. In this case, the exclusive policy issue can also be avoided. The experiments on the Mujoco and RoboSchool benchmarks demonstrate the advantages of the proposed policy.

**Strengths:**

1.	The paper is well-written and has good readability.
2.	The combination of bold actor learning and conservative critic learning seems interesting.
3.	Experiment results illustrate the advantages of the proposed method.

**Weaknesses:**

1.	In this paper, there are some assumptions that should be demonstrated by the theoretical analysis or the examples in real-world environments. (See the Questions part)
2.	Running time comparison should be given.
3.	The author did not explain the characteristics of the definition of DDQS. (See the Questions part)
4.	In the experiment, the numerical results should be given in different time steps.

**Questions:**

1.	I am confused about the assumption that policy learning lags behind Q-function learning, meaning that actions collected from the policy have relatively lower Q-values. There can be some theoretical analysis for this assumption.
2.	I am also confused about the assumption that the Q-function is multimodal. There can be some examples and experimental results.
3.	This paper supposes that the high Q-value actions far from the policy can be sampled. How can we get the “high Q-value actions” that are far from the current policy in policy learning?  There may be some noises or guided exploration functions for policy learning as in TD3 and SAC. These operations will not make actions far from the current policy.
4.	Why DDQS works? It seems that DDQS will select the actions that have the high values of $Q^{max}(s,a)$ and $Q^{min}(s,a)$. Why DDQS is better than other definitions based on $Q^{max}(s,a)$ and $Q^{min}(s,a)$, such as $\sum_{a}(Q^{max}(s,a)$+ $Q^{min}(s,a))/ \sum_{a}Q^{max}(s,a)$.

---

> ### Author Response · Authors · 2023-11-16
>
> Thank you for your review.
>
> **For question 1,**
>
>   1. Policy learning lags behind Q-function learning. From an empirical viewpoint, as described in our algorithm and previous algorithms (SAC, TD3), policy learning occurs after Q-function learning.
>   2. From a theoretical viewpoint, you can refer to the Soft Policy Iteration theorem in SAC, which mandates the initial Q update (soft policy evaluation) followed by policy update (soft policy improvement). In this scheme, Only when the Q value converges can the policy converge to the optimum.
>
> **For question 2,**
>
> &nbsp;&nbsp;&nbsp;&nbsp;In the maximum entropy RL framework, the optimal Q value should be multimodal. This assumption was first proposed in the SQL[1] paper, and we also verify this assumption in Appendix E.5 (page 20, Fig. 13(c)), which shows that the Q function is multimodal in practice.
>
>
> **For question 3,**
>
> &nbsp;&nbsp;&nbsp;&nbsp;We sample actions in the action space and then construct a new exploration policy with these actions corresponding to Q values. This operation weakens the relation between action and policy but strengthens the relation between action and Q value.
>
> **For qusetion 4,**
>
>   1. DDQS samples actions with high Q-values according to the Qmax function and updates the target Q-value using the Qmin value. Not only does this guarantee convergence of the two Q-values in theory, but in practical applications, for a specific state, the $Q^{max}$ function encourages the sampling of actions with high Q-values, while the $Q^{min}$ function ensures stable value updates. These two functions balance the learning of the two Q-functions.
>   2.  As for the new formula, we can simplify it to $(\sum_a (Q_1(s,a)+Q_2(s,a)))/Z$, since $Q^{max}$ and $Q^{min}$ are always different, and the partition function Z normalizes the distribution. If both $Q_{1}$ and $Q_{2}$ are large, this operator equals $Q^{max}$. However, if $Q_{1}(s,a)$ is large and $Q_{2}$ is small, the sum leads to an uniform sampling, which makes no sense.
>
> **For weakness 2,**
>
> &nbsp;&nbsp;&nbsp;&nbsp;We have discussed the time consumption in Appendix E.4, Page 16.
>
> **For weakness 4,**
>
> &nbsp;&nbsp;&nbsp;&nbsp;We have updated the PDF file. You can refer to Table F on Page 21 to check the numerical results. In different environments, the optimal results for these three time steps are all displayed in bold.
>
> [1]Haarnoja T, et al. Reinforcement learning with deep energy-based policies,ICML, 2017.
>
>
> Hope these comments address your concerns.

---

> > ### Comment · Reviewer_vPPz · 2023-11-17
> >
> > Thank you, authors for answering my questions and the rebuttal.
> > Most of my concerns are addressed.

---

### Official Review · Reviewer_3zdT · 2023-10-29

**Soundness:** 2 fair
**Presentation:** 2 fair
**Contribution:** 2 fair
**Rating:** 5
**Confidence:** 3

**Summary:**

The paper proposes an exploration method BACC for off-policy reinforcement learning. The method replaces the conventional blind action perturbation and exclusive policy with a Q-value-guided exploration scheme. The intuition is that a well-learned Q function could guide out-of-distribution exploration beyond the current policy. The method is instantiated under the double-Q framework. The major difference between the proposed method and the conventional double Q learning is in the exploration policy. In this work, the exploration policy distribution is estimated using the exponential of the maximum of Q functions. Experiments conducted on simulated continuous control tasks show slightly better performance of BACC than the baselines.

**Strengths:**

1. Enhancing exploration using guidance from value networks to jump out of the sub-optimality of the policy is a reasonable idea.
2. Motivating the BACC algorithm from the dynamic Boltzmann softmax update theorem and grounding the algorithm into double-Q learning sounds novel to me.

**Weaknesses:**

1. The paper is difficult to follow especially when reading Section 4. I actually cannot understand how all the introduced components fit together into the proposed method until reading Algorithm 1. I think it would be better to explicitly state how the DDQS operator is instantiated in Sec 4.2 and 4.3 for the readers to understand the method more easily. Also, I am still confused about why the objectives for Q and $\pi$ learning are induced using the conservative policy $\pi_O$; it does not match the implementation where the transitions are obtained using the exploration policy distribution $\pi_E$.
2. The experimented tasks are general RL testbeds, which may not be the best fit to verify the exploration ability of different methods. I would recommend that the authors experiment on sparse-reward control tasks, such as ``FetchPush'' from OpenAI Gym robotics suite.
3. The related literature of this paper is not sufficiently discussed. In the related work (Sec 4.5), many general-purpose exploration methods (not designed for specific RL algorithms) such as exploration with intrinsic motivation [1][2], exploration with randomized networks [3][4], explicitly separating out exploration [5] are not thoroughly discussed.

[1] Machado, Marlos C., Marc G. Bellemare, and Michael Bowling. "Count-based exploration with the successor representation." Proceedings of the AAAI Conference on Artificial Intelligence. Vol. 34. No. 04. 2020.

[2] Pathak, Deepak, Dhiraj Gandhi, and Abhinav Gupta. "Self-supervised exploration via disagreement." International conference on machine learning. PMLR, 2019.

[3] Burda, Yuri, et al. "Exploration by random network distillation." arXiv preprint arXiv:1810.12894 (2018).

[4] Fortunato, Meire, et al. "Noisy networks for exploration." arXiv preprint arXiv:1706.10295 (2017).

[5] Ecoffet, Adrien, et al. "First return, then explore." Nature 590.7847 (2021): 580-586.

4. The BACC algorithm is similar to the ``hybrid actor-critic exploration'' technique in AW-OPT[6], although it is motivated from a more practical aspect. I recommend the authors compare with or at least discuss this related technique.

[6] Lu, Yao, et al. "Aw-opt: Learning robotic skills with imitation and reinforcement at scale." Conference on Robot Learning. PMLR, 2022.

5. The empirical results seem not significant. In Fig. 3, the performance of BACC is close to the baselines except in ``Humanoid-v2''. In Fig. 5(a), the average performance of BACC is better but the big variance of the baseline OAC covers the confidence range of BACC, while in Fig. 5(b) BACC fails to gain advantage over SAC. It might be due to the benchmark tasks being too easy (dense reward, the learning agent could make progress even without proper exploration) and not favorable to methods with strong exploration abilities. Therefore, the authors are encouraged to investigate other tasks with stricter requirements for exploration.

**Questions:**

Please refer to the ``weaknesses'' part.

---

> ### Author Response · Authors · 2023-11-18
>
> Thanks for your review.
>
> **For weakness 1,**
>
>   1. We have made every effort to consider the coherence of paragraphs to ensure the continuity of the reading. For example, at the beginning and end of section 4.2, we attempted to connect sections 4.1 and 4.3.
>
>   2. As for the issue of mismatch:
>      * for the policy update, there appears to be no issue with the mismatch between the behavior policy and the target policy in off-policy algorithms. We explicitly introduce the conservative policy to establish a connection between the minimum Q value and policy learning. This provides a direct understanding of the role that the minimum Q value plays in the previous paper.
>      * for value update, we update the target value with transition sampled from the buffer(collected with policy $\pi_{E}$).
>
>
> **For weakness 2 and 3,**
>
>   1. We respectfully disagree with some of the viewpoints you've presented. Addressing sparse rewards indeed requires exploration methods based on the state space, but methods for enhancing exploration are not necessarily designed to solve the sparse reward problem.
>
>   2. Although our title includes exploration, it does not address sparse reward tasks (explore the state space). Instead, it focuses on achieving efficient continuous control in a complex action space(explore the action space).
>
>   3. Citations 1-5 are interesting papers, but they are not very relevant to our research content(problems descirbed in section 3).
>
>
> **for weakness 4,**
>
>   1. Thanks for introducing the AW-OPT paper, we also read the QT-OPT paper. We understand why you are concerned about the sparse reward issue.
>
>   2. The AW-OPT method and our approach do share some similarities. We have updated the paper PDF and introduced this work in the related work section.
>
>
> **for weakness 5,**
>
>   1. Figure 4 adequately illustrates that our proposed method can address the issues mentioned in Section 3. We present more comprehensive results not only to demonstrate the effectiveness of the approach but also to highlight its limitations.
>
>   2. From Fig.3(f) to Fig.3(a), the action space of the environment becomes increasingly complex. To demonstrate the effectiveness of our method, the performance in the challenging task (humanoid) should be more convincing. In Table F(appendix, page21), we present numerical results, showing that even if our results are not on par with the optimal ones, the numerical difference is minimal. Our method was not intentionally tuned.
>
>
>   3. Regarding the results in Fig.5 (a-b), in the flagrun environment, the changing flag position makes it challenging for the Q function to learn. Using the Q function to guide exploration in the early stages can lead to a performance loss, but this can be compensated for later (as indicated by the increasing trends in bacc and oac). The FlagrunHarder environment poses even greater difficulty. Please note the scale of the y-axis, where it's challenging for the episode return to reach 200, indicating that interaction occurs before the first flag change. In this scenario, both bacc and oac prove effective. The high variance in OAC is due to its difficulty in accurately estimating the exploration direction. It also reflects the robustness of our method, as we have mentioned in the paper.
>
> Hope this comment could address your concerns.

---

> > ### Comment · Reviewer_3zdT · 2023-11-21
> >
> > Dear authors,
> > Thank you for your efforts in the rebuttal.
> > My concerns in weakness 1 and 4 are addressed.
> > As for ``exploration’’ in state space or in action space, I would like to thank the authors for pointing out the distinction, but I still think that they both belong to exploration methods and the paper should discuss how BACC performs compared with those state-space exploration ones, possibly using the same dense-reward benchmark in this paper.
> > I agree that solving sparse-reward problems oftentimes requires more techniques besides exploration. I am curious to know whether BACC combined with relabeling tricks such as Hindsight Experience Replay that densify the reward signal can be applied to sparse-reward tasks.
> > Regarding the performance of BACC, I appreciate your detailed and frank explanation. However, the limited improvement in the final return of the trained policy remains a weakness.

---

> ### Author Response · Authors · 2023-11-21
>
> Dear Reviewer 3zdT,
>
> Thank you for your reply. We would like to talk more about sparse reward, even though it is not very related to the issue our paper aims to address.
>
>   1. We feel it is inappropriate to directly compare our method with sparse reward methods.
>
>      * Generally, state space exploration methods often involve generating intrinsic rewards (no reward is an extreme case of sparse reward). Applying such methods directly in dense reward environments results in reward conflicts. Deciding whether to depend on intrinsic rewards or environmental rewards for learning becomes a challenging task.
>      * Consider the noisy-TV problem[1], where intrinsic rewards impede the maximization of environmental rewards, presenting a challenge. And subsequent method[2] merely aim to alleviate reward conflicts.
>      * Currently, we lack a clear strategy for balancing intrinsic and environmental rewards when implementing sparse reward methods in dense reward environments. When the intrinsic reward is small, its necessity becomes questionable. On the other hand, if it is substantial, it intervenes the normal policy learning.
>
>   2.  (BACC+HER) This is indeed an interesting question. There seems to be a way to avoid conflicts very well. We could explore the action space to achieve the generated (dense reward) internal goal with BACC at a low level. Then, we can explore internal goals (state) with HER to achieve external goal at a higher level. This collaboration appears to be an ideal strategy.
>
> [1] Burda Y, et al. Exploration by random network distillation. ICLR, 2019.
>
> [2] Chen E, et al. Redeeming intrinsic rewards via constrained optimization. NIPS, 2022
>
> Hope this reply could address your concerns in weakness 2 and 3.

---

### Official Review · Reviewer_4MmP · 2023-10-31

**Soundness:** 3 good
**Presentation:** 3 good
**Contribution:** 2 fair
**Rating:** 5
**Confidence:** 3

**Summary:**

Exploring continuous action spaces can be difficult due to the vast number of possible actions, often leading to heuristic approaches. Previous research has shown the benefits of policy-based exploration in deterministic policy RL, but this approach can have issues such as unguided exploration and exclusive policy. To address these challenges, the Bold Actor Conservative Critic (BACC) approach uses Q-value to guide exploration and derive an exploration policy. This approach is evaluated on Mujoco and Roboschool benchmarks.

**Strengths:**

This paper studies how to improve exploration in continuous control in RL, which is a core topic in the realm of RL. The paper is also clearly written and easy to follow.

**Weaknesses:**

My main concern for the paper is the experimental evaluation part, where the proposed BACC method does not show significant performance improvement. In addition, as the title of the paper is "Careful at Estimation and Bold at Exploration for Deterministic Policy Gradient Algorithm", it would be better to investigate how well the value function estimates by studying the estimation error.

**Questions:**

> As shown in Figure 3, our method achieves promising results on this benchmark.

Indeed, the gap is quite marginal. In addition, what is the interval you used for the moving average? The plots seems to be too smooth, and doe not show clear benefits empirically.

> When evaluating the quality of exploration solely based on the results in Fig. 3, it might not provide a clear understanding of what is happening.

Can the authors better explain how this is done? Is it evaluated purely by the exploration method and do not use any kind of exploitation? In addition, it should compare with other exploration methods, e.g., random network distillation. Otherwise, the claim for improved exploration is not well-supported.

---

> ### Author Response · Authors · 2023-11-15
>
> Thank you for taking the time to review. We would like to provide additional explanations to address your concerns.
>
>  **For Strengths:**
>
> &nbsp;&nbsp;&nbsp;&nbsp;Thank you for recognizing our work in exploring the continuous action space. In practical applications, we believe this can be helpful in training robots to acquire some fundamental capabilities.
>
>  **For Weaknesses:**
>
> 1. Figure 4 illustrates the effectiveness of our approach in addressing the challenges presented in Section 3. And in Figure 3, the performance in the challenging task, particularly with the humanoid, is expected to be more convincing. From Fig. 3(f) to Fig. 3(a), the action space of the environment becomes progressively more complex. We present more comprehensive results not only to showcase the effectiveness of the approach but also to emphasize its limitations.
>
> 2. The "estimation error" has been thoroughly addressed in the TD3 paper. They update the target Q value by utilizing the minimum value of the double Q function to handle the impractical argmax operator in the continuous action space. In terms of value update, our approach is also grounded in this concept. We further propose the conservative policy to offer a direct understanding of the role that the minimum Q value plays in the previous paper.
>
>
> **For Questions,**
>
> 1. The interval is 100, and this interval remains consistent across all plots. In some of the plots, there is a noticeable amount of jaggedness in the lines.
>
> 2. "When evaluating the quality...," this primarily pertains to the results in Figure 4.
>     * What we want to convey is that it is not very intuitive to identify where the problem lies in the algorithm from the evaluation results. Instead, we should observe the exploration results to determine what happened.
>     * In Figure 3, the y-axis shows the evaluated average episode return(deterministic policy). In Figure 4, the y-axis displays the exploration average episode return(exploration policy).
>
> 3. The sparse reward is not the core issue that our paper aims to address. The problems we are addressing are outlined in Section 3, unguided exploration and exclusive policy.
>     * Although exploration(with more techniques) is the most effective way to tackle sparse rewards , it is not designed solely to address the sparse reward problem, it has wider applications.
>     * Our method explores the **action space** in **dense reward** environments to achieve more **efficient** robot control.
>
> 4. "compare with RND...",
>     * Sparse reward methods are not applicable to dense reward environments. It is challenging to balance environmental rewards and the intrinsic reward generated by the sparse reward exploration method. The most well-known issue is the Noisy-TV problem mentioned in the RND paper.
>     * This problem has also been further discussed in the following interesting paper [1].
>
>     [1] Chen E, et al. Redeeming intrinsic rewards via constrained optimization. NIPS, 2022
>
> Hope this comment could address your concerns.

---

### Meta-Review · Area_Chair_Z1LP · 2023-12-04

**Metareview:**

This paper presents a novel method to guide out-of-distribution exploration in actor-critic algorithms by leveraging the action-value function.

**Reviewers have reported the following strengths:**
- Importance of the studied topic;
- Significance and novelty of the idea;
- Good quality of writing.

**Reviewers have reported the following weaknesses:**
- Empirical evaluation;
- Theoretical motivation;
- Description of related works.

**Decision**

The authors' rebuttal helped solve the Reviewers' doubts. However, concerns remained about the empirical evaluation of this work. In particular, a major point of discussion was about the experimental setup, and its lack of focus on exploration problems. I understand the Reviewers' point that exploration is particularly interesting for sparse reward setting, but I disagree that this paper should have necessarily considered such setting. I deem the considered environments potentially enough. However, experimental results are not convincing. The performance improvement is statistically negligible compared to baselines in many problems. Together with the absence of strong theoretical motivation for the presented work, the lack of strong empirical evidence is a major issue with this paper.

I strongly encourage the authors to show more convincing results, and possibly, change the presentation of their work in order to avoid misunderstandings, as happened for this submission.

**Justification For Why Not Higher Score:**

The empirical evaluation and motivation of the paper remain major weaknesses that need to be addressed.

**Justification For Why Not Lower Score:**

N/A

---

### Decision · Program_Chairs · 2024-01-16

Reject